# Beyond Minimax: Structure-Aware Learning for Differential Games

## Abstract

A central challenge in artificial intelligence is to design agents that solve structured engineering problems, such as zero-sum differential games, without handcrafted solutions or expert demonstrations. Differential games capture multi-agent interactions with opposing objectives, where optimal strategies are defined by equilibrium conditions. Classical theory based on Pontryagin's Maximum Principle (PMP) and the Hamilton–Jacobi–Isaacs (HJI) equations provides principled foundations, but these conditions are rarely tractable in practice. Deep learning, by contrast, offers flexible function approximation but typically ignores such structure and depends on large datasets or extensive online interactions.

We introduce a framework that embeds equilibrium conditions and terminal constraints from the calculus of variations directly into the training objective. This enables neural networks to jointly learn state, control, and costate trajectories while handling variable terminal times and manifold-constrained terminal states, yielding approximate saddle-point equilibria. We illustrate our approach with the pursuit–evasion game *Lady in the Lake*, showing that our method recovers structural properties of analytical solutions and generalizes to novel scenarios without supervision, pointing toward principled, structure-aware deep models for solving previously intractable differential games.

## 1 Introduction

How can we build multi-agent AI systems that solve differential games with known dynamics but without access to equilibrium trajectories or reward functions? We focus on pursuit–evasion, a prominent class of zero-sum differential games where a pursuer aims to capture an evader actively seeking to avoid interception. Solving such problems requires computing optimal strategies for both players, typically within a two-player zero-sum framework. In particular, we study the *Lady in the Lake* game, a nonlinear system with a variable time horizon and no explicit reward function. This problem captures essential challenges of multi-agent decision-making and has applications in maritime collision avoidance, aeronautics, and security (Isaacs, 1965; Başar & Olsder, 1998).

Reinforcement learning methods excel when reward functions are explicitly designed (Mnih et al., 2013; Silver et al., 2016) or when large offline datasets are available (Wang et al., 2017; Levine et al., 2018), but neither is available in this setting. In problems such as *Lady in the Lake*, there is effectively *no reward function*: there is no running cost, the terminal time is a stopping variable, and the game terminates only when the evader reaches the boundary. As a consequence, the horizon is unbounded and agents may act indefinitely without ever receiving a learning signal unless a terminal constraint is satisfied. While heuristic reward shaping is possible, it risks misalignment with the true equilibrium. A further challenge is that the value function is inherently discontinuous: singular surfaces induce nonsmooth transitions (Bernhard, 1977) that neural networks struggle to approximate. These difficulties are compounded by the structure of the optimal solution itself—a saddle-point equilibrium—where naive gradient-based minimax optimization in deep learning is known to diverge (Saxena & Cao, 2021; Barnett, 2018).

Differential game theory offers a complementary perspective: rather than modeling iterative exchanges where one agent's gain offsets another's loss, it focuses on equilibrium states in which no player can unilaterally improve their outcome. The framework provides both necessary and sufficient optimal conditions for such equilibria. The necessary conditions extend Pontryagin's Max-

imum Principle (PMP), while the Hamilton–Jacobi–Isaacs (HJI) equation characterizes sufficient conditions. For problems with variable horizons, the calculus of variations further enables analysis under terminal boundary constraints. Together, these tools yield saddle-point trajectories with variable horizons and terminal conditions. While these optimality conditions–PMP, HJI, and terminal boundary formulations–are tractable in canonical settings such as *Lady in the Lake*, they are rarely solvable in closed form for general differential games.

Recent advances in deep learning offer a promising alternative, but most approaches rely on data-driven approximations or reinforcement learning, often without incorporating the underlying structure of the dynamic optimization problem. In contrast, we propose a deep learning framework that learns to solve differential games not from data but instead by internalizing the same principles used by mathematicians and scientists. Our method embeds the optimality conditions from the calculus of variations and PMP directly into the training objective, allowing the network to learn state, control, and costate trajectories along with terminal time that satisfy these optimality conditions without requiring ground-truth control data, expert demonstrations, or reward functions.

To benchmark our approach, we use the classical pursuit-evasion problem known as the *Lady in the Lake* for which there is a known solution. Our results demonstrate that deep learning models can be systematically designed to solve differential games by embedding optimality conditions into both the model architecture and training process, enabling learning directly from foundational engineering principles, without relying on ground-truth data or analytical solutions. As mentioned, we validate our method on a problem with known analytical solution, the proposed framework is general and extensible to a broad class of differential games where analytical solutions are unknown or intractable.

**Our main contributions are:**

- We introduce a principled training framework that embeds the calculus of variation and PMP, enabling networks to learn saddle-point equilibrium strategies along with terminal time in a fully unsupervised manner, without ground-truth controls or expert demonstrations.

- We propose a coordinate transformation and objective reformulation to overcome discontinuities in angular representation by neural networks, facilitating the design of neural architectures and end-to-end training.

- We show through experiments on pursuit–evasion games that the learned strategies recover key structural properties of analytical equilibrium solutions.

## 2 BACKGROUND

We study the classical pursuit–evasion game with variable terminal time and terminal constraints, *Lady in the Lake*, where a shoreline-constrained pursuer aims to capture an evader restricted to a circular lake. The game is played in a unit disk ($R = 1$) representing the lake. With reference to Figure 1, the evader $E$ starts at radius $r_0$ and a relative angular separation $\theta_0$ from the pursuer $P$. $E$ moves at constant speed $\mu$, while $P$ begins on the boundary of the disk and moves tangentially. The game ends when $E$ reaches the boundary.

The game state is represented in relative polar coordinates $(r(t), \theta(t))$, where $r(t)$ denotes the evader's radial distance from the center, and $\theta(t)$ is the angle between the evader $E$ and the pursuer $P$. The controls are the pursuer's tangential velocity $u_1(t) \in [-1, 1]$ and the evader's heading angle $u_2(t) \in (-\pi, \pi]$.

The relative dynamics are given by:

$$\dot{r} = v_2 \cos(u_2)$$
$$\dot{\theta} = \frac{v_2 \sin(u_2)}{r} - u_1 \tag{1}$$

where $\mu$ is the speed of the evader. The game ends at the terminal time $t_f$, defined as the first instant when the evader reaches the perimeter of the lake, e.g., $t_f = \min\{T : r(T) = 1\}$

The functional cost is $J(r, \theta, u_1, u_2) = |\theta(t_f)|$ where $t_f$ is the *variable* final time. $E$ tries to maximize $J$ while $P$ tries to minimize $J$

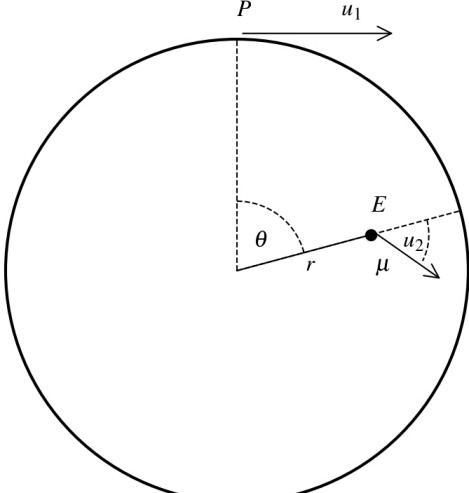

Figure 1: Illustration of the Lady in the Lake problem in relative polar coordinates. The evader $E$, located at $(r, \theta)$ inside the unit disk, moves at constant speed $\mu$ in direction $u_2$. The pursuer $P$ travels along the boundary at unit speed with heading $u_1$. In this relative coordinate system, the pursuer is fixed at the top of the circle, while the evader's motion is described relative to this reference frame.

$$\min_{u_1} \max_{u_2} \quad J(u_1, u_2)$$
$$\text{s.t.} \quad \text{dynamics in equation 1} \tag{2}$$
$$r(0) = r_0, \theta(0) = \theta_0, r(t_f) = 1$$

The equilibrium solution $(u_1^\star, u_2^\star)$ is the solution that satisfies

$$J(u_1^\star, u_2) \leq J(u_1^\star, u_2^\star) \leq J(u_1, u_2^\star) \quad \text{for all } u_1, u_2 \tag{3}$$

The solutions derived via Pontryagin's Maximum Principle (Moll & Pachter, 2024), and equivalently characterized by the Hamilton–Jacobi–Isaacs equation (Başar & Olsder, 1998), are given by:

$$u_1^\star(t) = \text{sgn}\big(\theta(t)\big), \qquad \sin u_2^\star(t) = \frac{\mu}{r(t)} \, \text{sgn}\big(\theta(t)\big). \tag{4}$$

The optimal strategy for the pursuer is to move at unit speed toward the evader along the smallest angular displacement, following a "bang–bang" policy characteristic of minimum-time optimal control. The evader, in turn, selects its motion so that the velocity $v_2$ in the real coordinate system remains perpendicular to the velocity $v_R$ in the relative polar coordinate system.

These saddle-point equilibrium strategies that avoid returning to the center are valid only within a specific domain, termed the *feasible region*, which is the focus of our experiments. The analysis of complementary regimes is deferred to future work.

The game exhibits a rich geometric structure: state dynamics evolve in polar coordinates, controls are bounded and discontinuous, and singular surfaces arise where backward dynamic programming fails, producing discontinuous value functions (Başar & Olsder, 1998). Additional challenges include the variable time horizon—the evader may remain in the lake for an arbitrarily long duration without penalty, though not indefinitely—and the absence of a running cost, with the objective specified solely at terminal time and subject to terminal state constraints (capture at the shoreline). These features induce discontinuities in value functions, nonconvex strategy spaces, and sharp transitions in optimal policies, making *Lady in the Lake* a demanding benchmark for both control-theoretic and reinforcement learning approaches.

## 3 RELATED WORK

We briefly review the relevant literature.

**Reinforcement Learning**: Several works have applied deep reinforcement learning to jointly train pursuers and evaders, with each agent optimizing its own reward (Qi et al., 2020; Xi & Cai, 2024; Xu et al., 2022; Wei et al., 2025). However, these approaches depend either on access to the true reward function—which is rarely available in engineering domains—or on carefully handcrafted surrogate rewards that can bias learning away from the original objective. In problems such as *Lady in the Lake*, there is effectively no reward function. Moreover, even when surrogate rewards are provided, the resulting value function is often irregular, since switching surfaces, capture regions, and dispersal lines introduce discontinuities that neural approximators struggle to represent. In contrast, our deep learning framework eliminates the need for reward specification by embedding calculus of variations and Pontryagin's Maximum Principle directly into the training loss, enabling agents to learn equilibrium strategies from first principles.

**Calculus of variations and optimal control.** The calculus of variations and optimal control derive necessary conditions for optimality by requiring that variations in a candidate trajectory vanish. This yields Pontryagin's Maximum Principle (PMP), together with boundary constraints. Differential game theory further extends these ideas through the Hamilton–Jacobi–Isaacs (HJI) equation, an analogue of dynamic programming for multi-agent settings. These formalisms have been applied to domains such as missile guidance, aircraft control, and pursuit–evasion scenarios (Isaacs, 1965; Başar & Olsder, 1998). However, solving PMP involves coupled state–costate boundary-value problems, while HJI-based approaches require nonlinear partial differential equations—both tractable only in low-dimensional or highly structured cases. This computational barrier has motivated approximate methods, including deep learning approaches that relax exact solutions while aiming to preserve underlying structure.

**Incorporating optimality conditions in neural networks.** Several works have embedded optimality conditions into neural architectures. Amos & Kolter (2017), Amos et al. (2018), and Donti et al. (2021) incorporate Karush–Kuhn–Tucker (KKT) conditions into constrained optimization layers, but focus on static decision variables. More recent approaches (Yin et al., 2024; Betti et al., 2024; Zhang et al., 2024) parameterize state and costate trajectories with neural networks, enforcing KKT and Pontryagin's Maximum Principle (PMP) conditions by rolling out and integrating the dynamics. While effective in fixed-horizon settings, these methods do not naturally extend to problems with variable terminal times or free terminal constraints, such as the *Lady in the Lake* game. By contrast, our approach leverages the calculus of variations to bypass fixed rollouts, enabling the learning of solutions with variable horizons and manifold-constrained terminal states while ensuring satisfaction of boundary conditions.

## 4 METHODOLOGY

We propose a framework, *Structure-Aware Learning for Differential Games (SAL-DG)*, that trains neural networks to approximate the state, costate, and control variables jointly using optimality conditions instead of data. We also propose reparameterization of the state and reformulation of the Lady in the Lake objective to make it learnable with this framework.

### 4.1 REPARAMETERIZATION OF THE STATE

Learning angular variables $\theta \in (-\pi, \pi]$ with neural networks is notoriously challenging due to discontinuities in their Euclidean representations (Zhou et al., 2018). In particular, the modulo-$2\pi$ mapping introduces artificial jumps that disrupt gradient-based optimization and undermine stable convergence during training. To address this, we reparameterize the state and reformulate the objective functional, enabling end-to-end training to remain effective and stable.

We design the state neural network $\hat{s}$ to output the state $\hat{s}(t) = \begin{bmatrix} \hat{r}(t) & \hat{x}(t) & \hat{y}(t) \end{bmatrix} \in \mathbb{R}^3$, where $r \in [0, 1]$, $(x, y)$ represents $(\sin\theta, \cos\theta)$ with the constraint $x^2 + y^2 = 1$ that enforces a valid angular embedding. Similarly, we design the costate neural network $\hat{\lambda}_s$ that outputs $\lambda_s = \begin{bmatrix} \hat{\lambda}_r & \hat{\lambda}_x & \hat{\lambda}_y \end{bmatrix} \in \mathbb{R}^3$.

The pursuer's control network $\mathcal{U}_1$ and the evader's control network $\mathcal{U}_2$ take as input the current state $s$ and costate $\lambda_s$, and output the pursuer's control $u_1 \in \mathbb{R}$ and the evader's control $u_e = (u_x, u_y)$. We parameterize the evader's action using an angle $u_2$, with $u_x = \sin u_2$ and $u_y = \cos u_2$, subject

to the constraint $u_x^2 + u_y^2 = 1$. This constraint is enforced through a normalization layer, ensuring that the output lies on the unit circle. The reformulated game dynamics are given by

$$
\begin{aligned}
\dot{r} &= \mu u_y, & \text{(denoted } g_r) \\
\dot{x} &= y\left(\tfrac{\mu u_x}{r} - u_1\right), & \text{(denoted } g_x) \\
\dot{y} &= -x\left(\tfrac{\mu u_x}{r} - u_1\right), & \text{(denoted } g_y).
\end{aligned} \tag{5}
$$

We propose to replace the original objective functional $J(u_1, u_2)$ with the surrogate functional

$$
\hat{J}(u_1, u_e) = -\cos\theta(t_f) = -y(t_f),
$$

The new optimization problem then becomes

$$
\begin{aligned}
\min_{u_1} \max_{u_e} \quad & \hat{J}(u_1, u_2) \\
\text{s.t.} \quad & \text{dynamics in equation 5} \\
& r(0) = r_0, x(0) = \sin(\theta_0), y(0) = \cos(\theta_0), r(t_f) = 1
\end{aligned} \tag{6}
$$

**Proposition 4.1** (Equivalence of Functionals). *The saddle-point solutions of equation 2 and equation 6 coincide; that is, both formulations admit the same equilibrium.*

*Proof sketch.* The function $\theta \mapsto -\cos\theta$ is an increasing function in the domain of $[0, \pi]$. The theta that maximizes (resp. minimizes) $|\theta|$ also maximizes (resp. minimizes) $-\cos\theta$ □

We define the Hamiltonian

$$
\mathcal{H}(s(t), \lambda_s(t), u_1(t), u_e(t), t) := \lambda_r g_r + \lambda_x g_x + \lambda_y g_y \tag{7}
$$

Necessary conditions for saddle-point equilibria are provided by the calculus of variations and Pontryagin's Maximum Principle (see Appendix A). Specifically,

$$
\dot{s^\star} = f(s^\star, u_1^\star, u_2^\star, \lambda_s^\star) \tag{8a}
$$

$$
\dot{\lambda_r^\star} = -\partial_r \mathcal{H}^\star, \quad -(\lambda_x^\star + \partial_x \mathcal{H})y^\star + (\lambda_y^\star + \partial_y \mathcal{H})x^\star = 0 \tag{8b}
$$

$$
u_1^\star = \arg\min_{u_1} \mathcal{H}(s^\star, u_1, u_2^\star, \lambda_s^\star) \tag{8c}
$$

$$
u_2^\star = \arg\max_{u_2} \mathcal{H}(s^\star, u_1^\star, u_2, \lambda_s^\star) \tag{8d}
$$

$$
s^\star(0) = s_0, r^\star(t_f) = 1 \tag{8e}
$$

$$
\lambda_x^\star(t_f)y^\star(t_f) + (-1 - \lambda_y^\star(t_f))x^\star(t_f) = 0 \tag{8f}
$$

## 4.2 TRAINING

This section outlines how to train state, costate, and control networks. This setup plays a critical role in the effectiveness of the proposed method, involving considerable subtle implementation challenges.

**Training control networks:** First, the control network is trained independently and state and costate are trained jointly. From equation 8c and equation 8d, and given the structure of the Hamiltonian $\mathcal{H}$, the equilibrium controls $u_1^\star$ and $u_2^\star$ are functions of the state and costate, i.e., $u_1^\star, u_2^\star = f(s^\star, \lambda_s^\star)$. The domains of these functions are thus restricted to the tuple $(s^\star, \lambda_s^\star)$.

We propose to model the control functions using neural networks $\mathcal{U}_1 : I \subset \mathbb{R}^6 \to \mathbb{R}$ and $\mathcal{U}_2 : I \subset \mathbb{R}^6 \to \mathbb{R}^2$, whose input domains $I$ include $(s^\star, \lambda_s^\star)$. The networks $\mathcal{U}_1$ and $\mathcal{U}_2$ are trained to minimize a loss function that enforces the necessary conditions for optimality derived from Pontryagin's Maximum Principle.

$$
\begin{aligned}
\text{Loss}_{\mathcal{U}_1} &= \mathbb{E}_{(s,\lambda_s)\sim\text{Uniform}(I)} \mathcal{H}(s, \lambda_s, \mathcal{U}_1(s, \lambda_s), u_{\text{dummy}}) \\
\text{Loss}_{\mathcal{U}_2} &= -\mathbb{E}_{(s,\lambda_s)\sim\text{Uniform}(I), u_1\sim D_1} \mathcal{H}(s, \lambda_s, u_{\text{dummy}}, \mathcal{U}_2(s, \lambda_s))
\end{aligned} \tag{9}
$$

This training paradigm is general and extends to other zero-sum differential games under the following assumption.

*Assumption* 1 (Time-autonomous and separable dependence). The Hamiltonian $\mathcal{H}(s(t), \lambda_s(t), u_1(t), u_e(t), t)$ is time-autonomous and admits separable dependence on the players' controls in the sense that its mixed second derivative vanishes, i.e.,

$$\frac{\partial \mathcal{H}}{\partial t} = 0, \quad \frac{\partial^2 \mathcal{H}}{\partial u_1 \, \partial u_2} = 0$$

This assumption implies that both agents select their actions at time $t$ independently, without explicit knowledge of the opponent's choice, relying only on the current state and costate. Consequently, each control can be learned as a function of state and costate alone. This independence enables $u_1$ and $u_2$ to optimize effectively while avoiding the difficulties of iterative minimax training; for instance, the evader can still learn meaningful strategies even when the pursuer is poorly trained.

**Training state, costate, and terminal time.** After training the control networks, we substitute $u_1, u_2$ in the Hamiltonian with their network outputs, i.e., functions of the state and costate. This mirrors the analytical elimination of variables in closed-form derivations. Instead of relying on ground-truth trajectories $s^\star$ and $\lambda_s^\star$, we learn approximations $\hat{s}$ and $\hat{\lambda}_s$ that minimize the residuals of the PMP equations (Raissi et al., 2019) in equation 8a and equation 8b.

$$\text{Loss}_{\text{PMP}} = \mathbb{E}_{t \sim U(0,T)} \left[ \|\dot{\hat{s}}(t) - f(\hat{s}(t), \hat{\lambda}_s(t), \mathcal{U}_1(\hat{s}(t), \hat{\lambda}_s(t)), \mathcal{U}_2(\hat{s}(t), \hat{\lambda}_s(t)))\|_2^2 + \|\dot{\hat{\lambda}}_s(t) - \partial_s \mathcal{H}\|_2^2 \right], \tag{10}$$

where $T$ is a heuristic time horizon chosen larger than the expected terminal time $t_f$, and $U(0, T)$ denotes the uniform distribution over $[0, T]$. This is key, or the model will fail to provide the correct solution.

From the boundary conditions implied by the calculus of variations, we also optimize $t_f$ through

$$\text{Loss}_{\text{B.C.}} = \|\hat{s}(0) - s_0\|_2^2 + (\hat{r}(t_f) - 1)^2 + \left( \hat{\lambda}_x(t_f) \hat{y}(t_f) + (-1 - \lambda_y(t_f)) \hat{x}(t_f) \right)^2 + \mathcal{H}(x)^2, \tag{11}$$

and define the total loss as

$$\text{Loss}_{\text{total}} = \alpha_1 \, \text{Loss}_{\text{PMP}} + \alpha_2 \, \text{Loss}_{\text{B.C.}}. \tag{12}$$

## 5 EXPERIMENTS

We evaluate different strategies on the *Lady in the Lake* pursuit–evasion game and compare their performance under consistent metrics.

### 5.1 EXPERIMENT SETUP AND METRICS

**Environment:** The game is played in a unit disk. The evader moves radially outward at constant speed $\mu = 0.25$, while the pursuer is restricted to the boundary. Dynamics follow equation 1 and equation 5. The evader starts at $r_0 = 0.3$ and $\theta_0 = \pi - 0.05$.

**Baseline:** We compare four strategies under identical state–action spaces: the analytic **Ground Truth** solution, our **SAL-DG** enforcing Pontryagin's Maximum Principle, and two reinforcement-learning baselines, **DDPG** and **TD3**.

**Evaluation Metrics:** We evaluate the different strategies using three complementary metrics that capture both qualitative and quantitative aspects of the pursuit–evasion game.

First, we visualize trajectories in real and relative coordinates to assess whether the trajectories match the true saddle-point equilibrium. Second, we analyze the pursuer's angular velocity, where the analytical solution prescribes a bang–bang policy with $u_1 \in \{-1, +1\}$. Learned policies are evaluated by whether their control profiles exhibit the same saturation behavior, and we quantify deviations from this optimal strategy using the mean squared error relative to the ideal bang–bang profile. Finally, we examine the angle between the evader's velocity $v_2$ and the relative velocity $v_R$. Equilibrium occurs when $\angle(v_2, v_R) = \frac{\pi}{2}$, and we measure deviations through the mean squared error from $\pi/2$, corresponding to the evader's optimal action.

## 5.2 RESULTS

**Trajectories:** A first point of comparison comes from overlaying the trajectories of the evader and pursuer across different strategies. Figure 2 shows the game in real coordinates: the analytic ground truth and SAL-DG closely follow the expected curved escape path, while the RL baselines exhibit outward motion but deviate from the optimal curvature. This indicates that shaping rewards encourage progress but fail to capture the geometric optimality of the analytic solution. Figure 3 further emphasizes this difference: in pursuer-fixed coordinates, the divergence of RL baselines becomes more pronounced, whereas SAL-DG remains consistent with the analytic trajectory. These comparisons provide qualitative evidence of SAL-DG's closer adherence to the differential-game equilibrium.

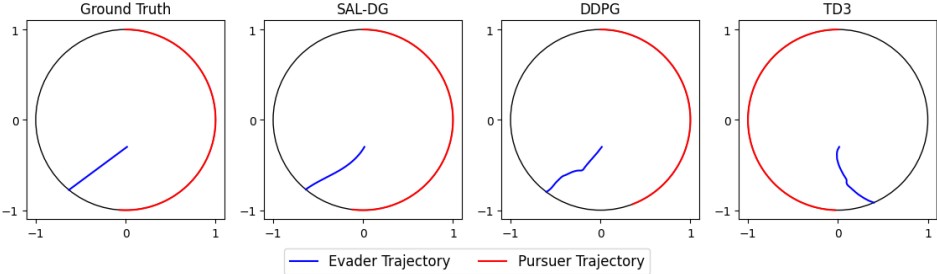

Figure 2: Comparison of evader and pursuer trajectories in real coordinates across different strategies (Ground Truth, SAL-DG, DDPG, and TD3). The analytic solution and SAL-DG follow the ground truth path, while the RL baselines deviate from the optimal curvature. For readability, only one representative trajectory is shown per strategy.

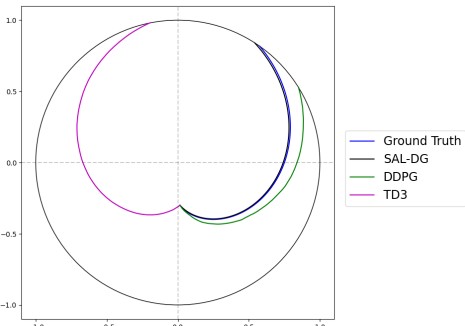

Figure 3: Comparison of evader and pursuer trajectories in relative coordinates across different strategies. The divergence between SAL-DG and the RL baselines becomes more pronounced in this representation. For readability, only one representative trajectory is shown per strategy.

**Pursuer's Control:** Figure 4(a) and Table 1 report the behavior of the pursuer's angular velocity magnitude $|u_1|$. SAL-DG achieves zero error, exactly matching the analytic saturation condition $|u_1| = 1$. By contrast, both DDPG and TD3 incur nonzero errors, and their profiles fluctuate below unit magnitude rather than maintaining it precisely. This indicates that while heuristic shaping rewards bias policies toward large control magnitudes, they fail to reproduce the analytic solution.

**Orthogonality between $v_2$ and $v_r$:** Figure 4(b) and Table 1 show the error relative to the orthogonality condition $\angle(v_2, v_r) = \pi/2$. SAL-DG again yields the lowest error, remaining closest to the analytic solution. RL baselines failed to maintain consistent orthogonality. Together, these results demonstrate that RL baselines do not reliably satisfy the equilibrium geometry.

## 5.3 SENSITIVITY TO REWARD

Reinforcement learning methods are highly sensitive to the design of reward functions. Balancing dense shaping terms against sparse terminal outcomes is non-trivial, and small changes in this

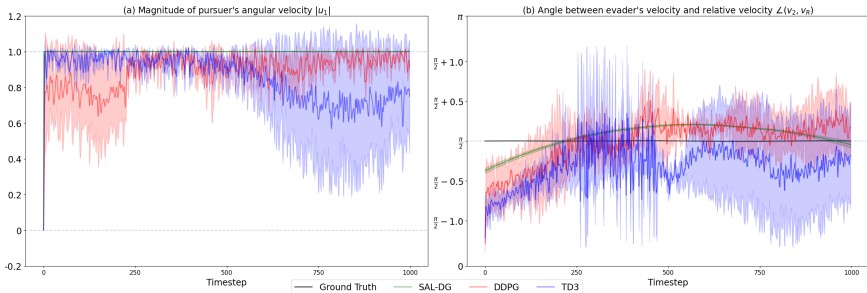

Figure 4: (a) Angular velocity profiles of the pursuer across different strategies. The analytic ground truth exhibits a bang–bang pattern with $u_1 \in \{-1, +1\}$, which is matched by SAL-DG. RL baselines are unable to encode bang–bang behavior. (b) Angle between evader's velocity and the relative velocity. SAL-DG yields lowest error among all strategies. Note that all strategies are shown across three random seeds.

balance can drastically alter learned strategies. To illustrate this, we vary the mixing parameter $\omega$ in

$$ r = \omega \cdot r_{\text{shaping}} + (1 - \omega) \cdot r_{\text{terminal}} $$

and evaluate the resulting policies. Table 1 summarizes the outcomes. The results show that both DDPG and TD3 are strongly affected by the choice of $\omega$. These inconsistencies highlight the fragility of heuristic reward design: RL baselines may capture outward progress but fail to reliably enforce the geometric equilibrium conditions. By contrast, SAL-DG avoids this sensitivity altogether. Trained directly from Hamiltonian dynamics under Pontryagin's Maximum Principle, it achieves zero error in estimated $u_1$ and the lowest error in orthogonality, without requiring any reward engineering. This demonstrates the advantage of structure-preserving training over trial-and-error reward specification.

| Mixing $\omega$ | MSE for $|u_1|$ | | MSE for $\angle(v_2, v_R)$ | |
|---|---|---|---|---|
| | DDPG | TD3 | DDPG | TD3 |
| 0.25 | $0.025 \pm 0.014^*$ | $0.110 \pm 0.133$ | $0.684 \pm 0.368$ | $1.246 \pm 0.695$ |
| 0.50 | $0.035 \pm 0.028$ | $0.078 \pm 0.099^*$ | $0.155 \pm 0.109^*$ | $0.380 \pm 0.227$ |
| 0.75 | $0.202 \pm 0.122$ | $0.079 \pm 0.051$ | $1.083 \pm 0.648$ | $0.237 \pm 0.113^*$ |
| SAL-DG | $\mathbf{0.000 \pm 0.000}$ | | $\mathbf{0.026 \pm 0.002}$ | |

Table 1: Reward sensitivity to the mixing parameter $\omega$ for DDPG and TD3, reporting MSE of control magnitude $|u_1|$ and orthogonality $\angle(v_2, v_r)$. SAL-DG does not require reward mixing and achieves zero error by construction. (An asterisk (*) indicates the best-performing $\omega$ for each RL baseline in a given column.)

## 5.4 Ablation Study: Control-Based Reward

A key benefit of our reformulation of the Lady in the Lake problem is that the surrogate terminal cost is differentiable, allowing it to move inside the integral.

$$ \hat{J} = -y(T) = -y(0) + \int_0^T -\dot{y}(t)\, dt, \qquad \dot{y}(t) = -x(t)\left( \frac{\mu u_x(t)}{r(t)} - u_1(t) \right). $$

This motivates a control-based reward defined directly from $-\dot{y}(t)$, so that the cumulative return is equivalent (up to constants) to the terminal objective. Unlike the heuristic reward, this formulation introduces no auxiliary shaping terms and aligns precisely with the underlying optimal-control problem. Figure 5 compares trajectories from the ground truth, SAL-DG, and reinforcement learning (RL) strategies trained with the control-based reward. Under this formulation, DDPG learns an evader strategy that moves toward the center and remains there. This behavior arises because approaching the center maximizes the accumulated running cost, while no penalty is imposed for time. Consequently, the evader never reaches the perimeter and the game fails to terminate. To

enforce termination, the reward must be augmented with a time-dependent penalty when the evader has not reached the circle . However, this modification breaks fidelity to the original objective, and the resulting strategy deviates from the true equilibrium trajectory. In contrast, the calculus-of-variations approach requires no such reward engineering: it directly enforces the terminal condition (e.g., $r(t_f) = 1$) and consistently produces trajectories aligned with the analytical equilibrium.

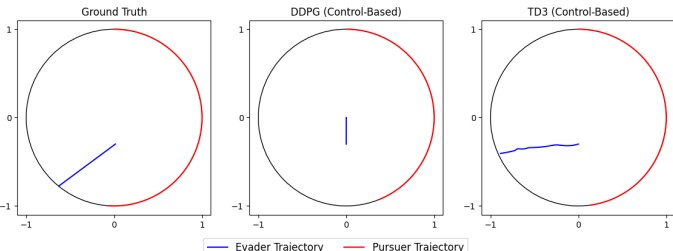

Figure 5: Trajectories of the evader (blue) and pursuer (red) under the **control-based** reward formulation. Although this reward is theoretically aligned with the terminal objective, RL baselines fail to reproduce the equilibrium strategy: DDPG collapses toward the center and stalls, while TD3 eventually reaches the boundary but along a distorted path. Maximizing the running cost encourages behavior inconsistent with the true objective.

## 6 Discussion and Future Work

We conclude by outlining the main limitations, scope, and directions for extension.

**Limitations:** Our framework enforces the necessary conditions of Pontryagin's Maximum Principle (PMP), which hold for open-loop trajectories rather than feedback policies. Synthesizing feedback requires converting open-loop solutions into policies, a step left for future work. We also assume known dynamics, whereas many real-world systems require learning them from data.

**Scope:** We restrict attention to the *feasible region* where equilibrium strategies exist. Outside this set—e.g., when the evader nears the center—division by $r$ causes numerical instabilities.

**Future Work:**

- **Learning dynamics:** Extending SAL-DG to settings with unknown or data-driven dynamics (Finn & Levine, 2017; Watter et al., 2015).
- **Feedback synthesis:** Training policies on generated state–costate trajectories to recover feedback laws.
- **Beyond feasible regions:** Handling trajectories outside the *feasible region*, such as near the center in Lady in the Lake.

These extensions point toward unifying principled control theory with scalable deep learning for multi-agent dynamic environments.

## 7 Conclusion

We present *Structure-Aware Learning for Differential Games (SAL-DG)*, a design paradigm for deep neural networks that learns from first principles rather than data. Our framework integrates the necessary conditions of the calculus of variations and PMP directly into the architecture and training objective, enabling networks to derive optimal strategies in differential games with *variable* time horizons and terminal constraints, without supervision or expert demonstrations. We validate this approach on the classical Lady in the Lake pursuit–evasion problem, where SAL-DG recovers known analytical solutions solely from problem specifications. More broadly, SAL-DG establishes a foundation for tackling higher-dimensional and analytically intractable games, offering a principled path toward interpretable and generalizable deep learning in multi-agent dynamic environments.

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

## A OPTIMALITY CONDITION

We consider the zero-sum game with a variable time horizon $[0, t_f]$, where the goal is to determine the control strategies

$$u_1^\star(t), \ u_e^\star(t), \qquad t \in [0, t_f],$$

that generate a valid saddle-point trajectory, i.e., a trajectory that satisfies the dynamics

$$\dot{s}^\star(t) = f(s, u_1, u_e)$$

and reaches the stopping set

$$\mathcal{S} = \{ s \mid r = 1 \}.$$

That is, the terminal condition satisfies $(s(t_f), t_f) \in \mathcal{S}$. The objective is to minimize–maximize the performance index

$$\hat{J}(s, u_1, u_e) = q_T\big(s(t_f), t_f\big).$$

The constrained optimization problem is therefore formulated as

$$
\begin{aligned}
\min_{u_1} \max_{u_e} \quad & \hat{J}(u_1, u_e) \\
\text{s.t.} \quad & \dot{s}(t) = f(s(t), u_1(t), u_e(t)), \quad t \in [0, t_f], \\
& r(0) = r_0, \quad x(0) = \sin(\theta_0), \quad y(0) = \cos(\theta_0), \\
& r(t_f) = 1.
\end{aligned}
\tag{13}
$$

To enforce the dynamic constraints, we introduce the Lagrange multipliers

$$\lambda_s(t) := \begin{bmatrix} \lambda_r(t) & \lambda_x(t) & \lambda_y(t) \end{bmatrix},$$

and define the augmented functional

$$\mathcal{J}_{\text{aug}}(s, \lambda_s, u_1, u_e, t_f) = \underbrace{-y(t_f)}_{\Phi(y(t_f), t_f)} + \int_0^{t_f} \left[ \lambda_r(g_r - \dot{r}) + \lambda_x(g_x - \dot{x}) + \lambda_y(g_y - \dot{y}) \right] dt. \quad (14)$$

**Proposition A.1.** *The valid equilibrium trajectories of the constrained optimization problem equation 13 and the augmented functional equation 14 coincide.*

*Proof sketch.* For any admissible trajectory that satisfies the dynamics, the residual terms inside the integral vanish, and the augmented functional reduces to the original objective equation 13. ☐

We now study equilibrium trajectories with respect to the augmented functional 14. Let $(s^\star, \lambda^\star, u_1^\star, u_e^\star)$ be the equilibrium trajectory. Note that an equilibrium trajectory $(s^\star, \lambda^\star, u_1^\star, u_e^\star)$ is defined only on the interval $[0, t_f^\star]$. Consider a perturbation of the state trajectory $s^\star$ by a *valid variation* $\delta s$, which modifies the terminal time to $t_f^\star + \delta t_f$ and the terminal state to

$$r^\star(t_f^\star) + \delta r(t_f^\star + \delta t_f), \quad x^\star(t_f^\star) + \delta x(t_f^\star + \delta t_f), \quad y^\star(t_f^\star) + \delta y(t_f^\star + \delta t_f).$$

To compute the variation, we evaluate the augmented functional along the perturbed trajectory

$$\left( s^\star + \delta s, \ \lambda_s^\star + \delta \lambda_s, \ u_1^\star + \delta u_1, \ u_e^\star + \delta u_e \right),$$

$$\mathcal{J}_{aug}(s^\star + \delta s, \lambda_s^\star + \delta \lambda, u_1^\star + \delta u_1, u_e^\star + \delta u_2)$$
$$= \Phi(y^\star(t_f) + \delta y(t_f^\star + \delta t_f), t_f^\star + \delta t_f)$$
$$+ \int_{t_f^\star}^{t_f^\star + \delta t} (\lambda_r + \delta \lambda_r)(g_r(s^\star + \delta s) - (r^\star \dot{+} \delta r)) + (\lambda_x + \delta \lambda_x)(g_x(s^\star + \delta s) - (x^\star \dot{+} \delta x))$$
$$+ (\lambda_y + \delta \lambda_y)(g_y(s^\star + \delta s) - (y^\star \dot{+} \delta y)) dt$$
$$+ \int_0^{t_f^\star} (\lambda_r^\star + \delta \lambda_r)(g_r(s^\star + \delta s) - (r^\star \dot{+} \delta r)) + (\lambda_x^\star + \delta \lambda_x)(g_x(s^\star + \delta s) - (x^\star \dot{+} \delta x))$$
$$+ (\lambda_y^\star + \delta \lambda_y)(g_y(s^\star + \delta s) - (y^\star \dot{+} \delta y)) dt$$

We calculate the variation in the equilibrium trajectory that is the linear terms of $\delta s$ in $\Delta \mathcal{J}_{aug} := \mathcal{J}_{aug}(s^\star + \delta s, \lambda_s^\star + \delta \lambda, u_1^\star + \delta u_1, u_e^\star + \delta u_2) - \mathcal{J}_{aug}(s^\star, \lambda_s^\star, u_1^\star, u_e^\star)$ First, we look at the terms in $\Delta \mathcal{J}$ that are linear in $\delta r, \delta \dot{r}$ in the integral from 0 to $t_f^\star$. There are

$$\int_0^{t_f^\star} \lambda_r^\star \frac{\partial g_r}{\partial r} \delta r + \lambda_x^\star \frac{\partial g_x}{\partial r} \delta r + \lambda_y^\star \frac{\partial g_y}{\partial r} \delta r - \lambda_r^\star \dot{\delta r} dt = [\lambda_r^\star \delta r]_0^{t^\star} + \int_0^{t_f^\star} \partial_r \mathcal{H} \delta r + \dot{\lambda}_r^\star dt$$

The linear terms of $\delta x, \delta y$ are the same. We can simplify the variation as

$$\delta \mathcal{J}_{aug}(s^\star, \lambda_s^\star) = \frac{d\Phi}{dy} \delta y_f + \left[ \mathcal{H}(t_f) - \lambda_r^\star(t_f) \dot{r^\star}(t_f) - \lambda_x^\star(t_f) \dot{x^\star}(t_f) - \lambda_y^\star(t_f) \dot{y^\star}(t_f) \right] \delta t_f$$
$$- \lambda_r^\star(t_f^\star) \delta r(t^\star) - \lambda_x^\star(t_f^\star) \delta x(t^\star) - \lambda_y^\star(t_f^\star) \delta y(t^\star)$$
$$+ \int_0^{t_f^\star} \left[ \dot{\lambda}_r + \partial_r \mathcal{H} \right] \delta r + \left[ \dot{\lambda}_x + \partial_x \mathcal{H} \right] \delta x + \left[ \dot{\lambda}_y + \partial_y \mathcal{H} \right] \delta y$$
$$+ \partial_{u_1} \mathcal{H} \delta u_1 + \partial_{u_e} \mathcal{H} \delta u_e + (g_r - \dot{r}) \delta \lambda_r + (g_x - \dot{x}) \delta \lambda_x + (g_y - \dot{y}) \delta \lambda_y dt$$

By approximation,

$$\delta r(t^\star) + \dot{r^\star}(t_f^\star)\delta t = (r^\star + \delta r)(t_f^\star) - r^\star(t_f^\star) + \dot{r^\star}(t^\star)\delta t$$
$$\approx (r^\star + \delta r)(t_f^\star) + (\dot{r^\star + \delta r})(t^\star)\delta t_f - r^\star(t_f^\star)$$
$$\approx (r^\star + \delta r)(t_f^\star) + \dot{\delta r}(t^\star)\delta t_f - r^\star(t_f^\star)$$
$$:= \delta r_f$$

Similarly,

$$\delta x(t^\star) + \dot{x^\star}(t_f^\star)\delta t_f = \delta x_f + o(||\delta x||, ||\delta t||)$$
$$\delta y(t^\star) + \dot{y^\star}(t_f^\star)\delta t_f = \delta y_f + o(||\delta y||, ||\delta t||)$$

The variation is simplified as

$$\delta \mathcal{J}_{aug}(s^\star, \lambda_s^\star) = \left[-\lambda_y^\star(t_f^\star) - 1\right]\delta y_f - \lambda_x^\star(t_f^\star)\delta x_f - \lambda_r^\star(t^\star)\delta r_f + \mathcal{H}(s^\star, \lambda_s^\star)\delta t_f$$

$$+ \int_0^{t_f^\star} \left[\dot{\lambda_r^\star} + \partial_r \mathcal{H}\right]\delta r + \left[\dot{\lambda_x^\star} + \partial_x \mathcal{H}\right]\delta x + \left[\dot{\lambda_y^\star} + \partial_y \mathcal{H}\right]\delta y$$

$$+ \partial_{u_1}\mathcal{H}\delta u_1 + \partial_{u_e}\mathcal{H}\delta u_e + (g_r - \dot{r})\delta\lambda_r + (g_x - \dot{x})\delta\lambda_x + (g_y - \dot{y})\delta\lambda_y$$
$$+ (\mathcal{H}(s^\star, \lambda_s^\star, u_1^\star + \delta u_1, u_e^\star) - \mathcal{H}(s^\star, \lambda_s^\star, u_1^\star, u_e^\star)$$
$$+ \mathcal{H}(s^\star, \lambda_s^\star, u_1^\star, u_e^\star + \delta u_e) - \mathcal{H}(s^\star, \lambda_s^\star, u_1^\star, u_2^\star) \, dt$$

First, since $s^\star$ satisfies the dynamics constraint,

$$(g_r - \dot{r^\star})\delta\lambda_r + (g_x - \dot{x^\star})\delta\lambda_x + (g_y - \dot{y^\star})\delta\lambda_y = 0$$

Next, we choose $\lambda_r^\star, \lambda_x^\star, \lambda_y^\star$ so that for all admissible $\delta r, \delta x, \delta y$, the terms are

$$\left[\dot{\lambda_r^\star} + \partial_r \mathcal{H}\right]\delta r + \left[\dot{\lambda_x^\star} + \partial_x \mathcal{H}\right]\delta x + \left[\dot{\lambda_y^\star} + \partial_y \mathcal{H}\right]\delta y$$

is 0. Since $\delta r$ can be arbitrary and $(\delta x, \delta y)$ must be tangent to the circle $x^2 + y^2 = 1$ at point $(x^\star, y^\star)$, i.e. $\left\langle \begin{bmatrix} x \\ y \end{bmatrix}, \begin{bmatrix} \delta x \\ \delta y \end{bmatrix} \right\rangle = 0$, we choose $\lambda_r^\star, \lambda_x^\star, \lambda_y^\star$ such that

$$\dot{\lambda_r^\star} + \partial_r \mathcal{H} = 0$$
$$\left\langle \begin{bmatrix} \dot{\lambda_x^\star} + \partial_x \mathcal{H} \\ \dot{\lambda_y^\star} + \partial_y \mathcal{H} \end{bmatrix}, \begin{bmatrix} -y \\ x \end{bmatrix} \right\rangle = 0$$
$$\left\langle \begin{bmatrix} \lambda_x^\star(t_f^\star) \\ \dot{\lambda_y^\star} + \partial_y \mathcal{H} \end{bmatrix}, \begin{bmatrix} -y^\star(t_f^\star) \\ x^\star(t_f^\star) \end{bmatrix} \right\rangle = 0 \tag{15}$$
$$\mathcal{H}(s^\star, \lambda_s^\star, u_1^\star, u_e^\star) = 0$$

Such lambda exists as a result of the following proposition

**Proposition A.2** (Existence of the lagrangian multipliers). *Let $r^\star(t), x^\star(t), y^\star(t), u_1^\star(t), u_2^\star(t)$ be bounded functions defined on the domain $[0, t_f^\star]$ and $q_T \in \mathbb{R}^3$. Define the matrix*

$$A(t) := \begin{bmatrix} \partial_r g_r & \partial_r g_x & \partial_r g_y \\ \partial_x g_r & \partial_x g_x & \partial_x g_y \\ \partial_y g_r & \partial_y g_x & \partial_y g_y \end{bmatrix}\Bigg|_{r^\star(t), x^\star(t), y^\star(t), u_1^\star(t), u_2^\star(t)}$$

*Suppose $A(t)$ is Riemann integrable, then for all integrable function $\nu(t)$ and the terminal state $\Lambda_T \in \mathbb{R}^3$, there exists a function $\lambda(t)$ satisfying the differential equation:*

$$\dot{\lambda}(t) + \partial_s \mathcal{H}(t) = \nu(t) \quad \forall t \in [0, t_f^\star], \lambda(t_f^\star) = \Lambda_T$$

*Proof.*

$$\begin{bmatrix} \dot{\lambda}_r(t) \\ \dot{\lambda}_x(t) \\ \dot{\lambda}_y(t) \end{bmatrix} + \begin{bmatrix} \lambda_r\partial_r g_r + \lambda_x\partial_r g_x + \lambda_y\partial_r g_y \\ \lambda_r\partial_x g_r + \lambda_x\partial_x g_x + \lambda_y\partial_x g_y \\ \lambda_r\partial_y g_r + \lambda_x\partial_y g_x + \lambda_y\partial_y g_y \end{bmatrix} = \eta(t),$$

which can be rewritten as

$$
\begin{bmatrix} \dot{\lambda}_r(t) \\ \dot{\lambda}_x(t) \\ \dot{\lambda}_y(t) \end{bmatrix} + \underbrace{\begin{bmatrix} \partial_r g_r & \partial_r g_x & \partial_r g_y \\ \partial_x g_r & \partial_x g_x & \partial_x g_y \\ \partial_y g_r & \partial_y g_x & \partial_y g_y \end{bmatrix}}_{A(t)} \begin{bmatrix} \lambda_r(t) \\ \lambda_x(t) \\ \lambda_y(t) \end{bmatrix} = \eta(t)
$$

The differential equation has the solution

$$
\lambda(t) = e^{-\int_{t_f^\star}^{t} A(s)ds} \Lambda_T + e^{-\int_{t_f^\star}^{t} A(s)ds} \int_{t_f}^{t} e^{\int_{t_f^\star}^{s} A(w)dw} \eta(s)ds
$$

$\square$

It suffices, for example, to choose

$$
\nu(t) = \begin{bmatrix} 0 \\ x^\star(t) \\ y^\star(t) \end{bmatrix}, \qquad \Lambda_T = \begin{bmatrix} -\dfrac{x^\star(t_f^\star)g_x}{g_r} - \dfrac{y^\star(t_f^\star)g_y}{g_r} \\ x^\star(t_f^\star) \\ y^\star(t_f^\star) \end{bmatrix},
$$

noting that $g_r$ at the terminal time is nonzero (the evader's radial speed must remain nonvanishing at the final instant). With this choice, one can explicitly construct $\lambda_s$ satisfying equation 15, thereby establishing existence.

The only remaining term in the variation is

$$
\int_0^{t_f^\star} \Big( \mathcal{H}(s^\star, \lambda_s^\star, u_1^\star + \delta u_1, u_e^\star) - \mathcal{H}(s^\star, \lambda_s^\star, u_1^\star, u_e^\star) + \mathcal{H}(s^\star, \lambda_s^\star, u_1^\star, u_e^\star + \delta u_e) - \mathcal{H}(s^\star, \lambda_s^\star, u_1^\star, u_e^\star) \Big) dt.
$$

At equilibrium, neither player can improve their outcome by unilaterally deviating (i.e., setting $\delta u_1 = 0$ or $\delta u_e = 0$). Fixing the opponent's strategy, each unilateral perturbation makes the augmented cost functional no better for the deviating player:

$$
\mathcal{J}_{\text{aug}}(u_1^\star + \delta u_1, u_e^\star) \leq \mathcal{J}_{\text{aug}}(u_1^\star, u_e^\star), \qquad \mathcal{J}_{\text{aug}}(u_1^\star, u_e^\star + \delta u_e) \geq \mathcal{J}_{\text{aug}}(u_1^\star, u_e^\star).
$$

This captures the saddle-point property: the pursuer minimizes while the evader maximizes. Consequently, the optimal strategies satisfy

$$
u_1^\star = \arg\min_{u_1 \in \mathcal{U}} \mathcal{H}(r^\star, x^\star, y^\star, u_1, u_e^\star),
$$

$$
u_e^\star = \arg\max_{u_e \in \mathcal{U}} \mathcal{H}(r^\star, x^\star, y^\star, u_1^\star, u_e).
$$

**Remarks.** The terminal conditions could also be obtained by introducing additional Lagrange multipliers: $\lambda_d(t)$ to enforce the path constraint $x(t)^2 + y(t)^2 = 1$, and $\lambda_R$ to enforce the terminal constraint $x_f^2 + y_f^2 = 1$, followed by an application of PMP. This leads to conditions of the form

$$
\lambda_x^\star + \partial_x \mathcal{H} + 2\lambda_d x = 0, \qquad \lambda_y^\star + \partial_y \mathcal{H} + 2\lambda_d y = 0.
$$

To avoid introducing additional multipliers—which would enlarge the set of variables to be optimized by the neural network—we instead adopt a geometric argument and replace these expressions with the compact formulation in equation 15.

## B  HEURISTIC REWARD

The heuristic reward is implemented as a convex combination of shaping and terminal components:

$$r = w \cdot r_{\text{shaping}} + (1 - w) \cdot r_{\text{terminal}}, \tag{16}$$

where $w \in [0, 1]$ balances dense per-step signals with sparse outcome signals.

For the evader, the shaping reward encourages outward motion, angular separation, and urgency:

$$r_{\text{evader}}^{\text{shaping}} = \alpha_1 \cdot \dot{r} + \alpha_2 \cdot \theta - \alpha_3 \cdot (1 - r) + \alpha_4,$$

where $\dot{r}$ is the radial velocity, $\theta$ is the angular separation between evader and pursuer, and $(1 - r)$ penalizes slow progress toward the boundary. The terminal reward enforces outcome-driven signals:

$$r_{\text{evader}}^{\text{terminal}} = \begin{cases} +K_\theta \theta(T), & \text{if escaped,} \\ -K_{\text{cap}}, & \text{if captured,} \\ f(r^2\theta), & \text{if timeout,} \end{cases}$$

where $r$ is the evader's radial position and $T$ is the terminal time.

For the pursuer, the shaping reward encourages reducing angular separation, moving in the correct direction, and maintaining activity:

$$r_{\text{pursuer}}^{\text{shaping}} = \beta_1 \cdot (\pi - \theta) - \beta_2 \cdot \Delta\theta + \beta_3 \cdot |\omega| + \beta_4 \cdot d_{\text{dir}} + \beta_5 \cdot \frac{\pi - \theta}{\pi},$$

where $\Delta\theta$ is the change in separation angle, $\omega$ is the pursuer's angular velocity, and $d_{\text{dir}} \in \{\pm 1\}$ indicates whether the pursuer moves in the correct direction toward the evader. The terminal reward mirrors the evader's:

$$r_{\text{pursuer}}^{\text{terminal}} = \begin{cases} +K_{\text{cap}}, & \text{if captured,} \\ -K_\theta \theta(T), & \text{if evader escaped,} \\ g(r^2\theta), & \text{if timeout.} \end{cases}$$

Thus, $r$ is the evader's radial position, $\theta$ is the angular separation, $\dot{r}$ is radial velocity, $\Delta\theta$ is angular change, $\omega$ is the pursuer's angular velocity, and $d_{\text{dir}}$ encodes directional correctness.

All coefficients $\alpha_i, \beta_i$ and constants $K_\theta, K_{\text{cap}}$ denote fixed scalar weights.

This design ensures that both agents receive continuous shaping feedback during play while still being dominated by escape or capture outcomes.

## C  TRAJECTORIES

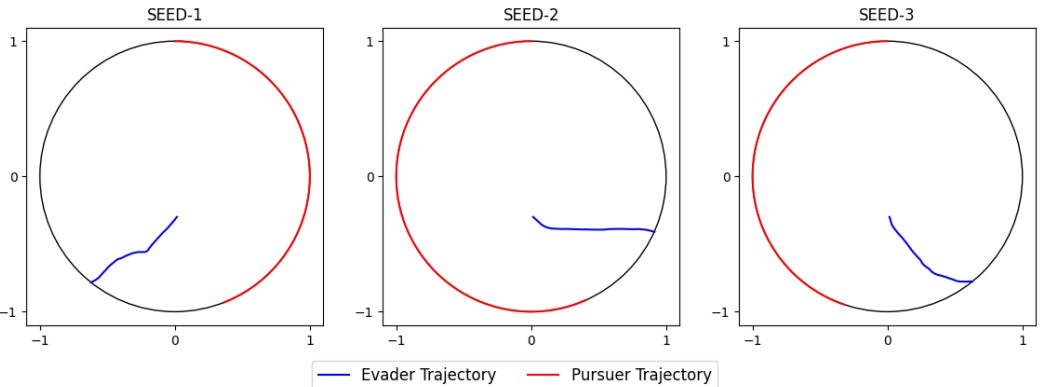

Figure 6: DDPG trajectories across three random seeds.

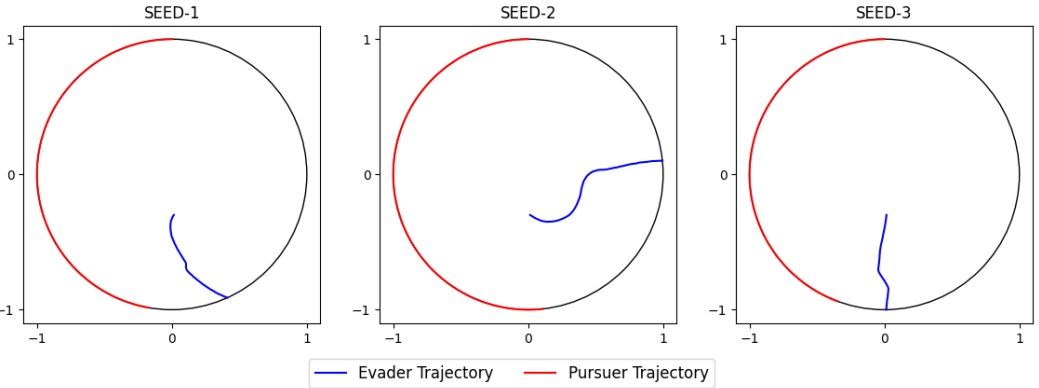

Figure 7: TD3 trajectories across three random seeds.

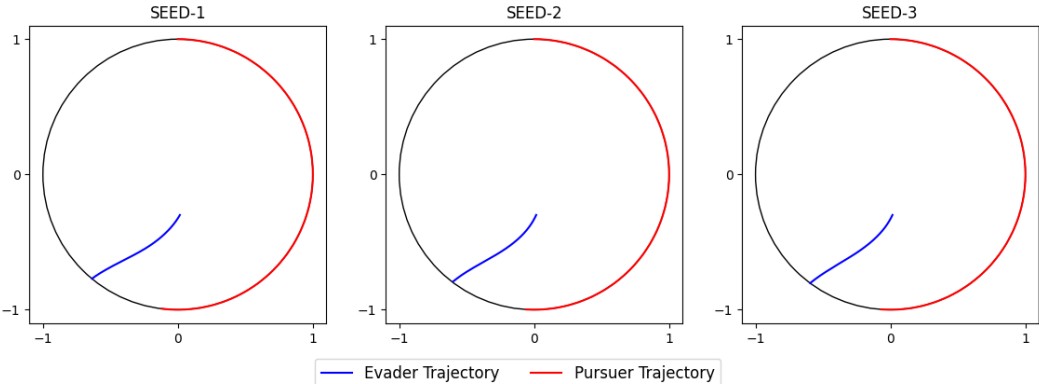

Figure 8: SAL-DG trajectories across three random seeds.

