# OpenReview forum: "Beyond Minimax: Structure-Aware Learning for Differential Games"
_ICLR.cc/2026/Conference — Submitted to ICLR 2026_

### Official Review · Reviewer_tkW5 · 2025-10-20

**Soundness:** 2
**Presentation:** 3
**Contribution:** 1
**Rating:** 2
**Confidence:** 3

**Summary:**

The paper investigates a particular class of differential games, namely a free-end time differential game with zero-sum payoff. The primary motivation is to develop a scalable solution technique to solve this game, which can get intractable when solved using classical techniques based on Pontryagin’s Maximum Principle (PMP). By embedding optimality conditions from calculus of variations and PMP into the training objective, the authors train neural networks to predict the state, control, co-state trajectories, and the terminal time, without any ground-truth data, or demonstrations.

**Strengths:**

Good presentation.

**Weaknesses:**

The paper provides marginal contribution at best, primarily because there are number of papers that solve 2p0s differential games using physics-informed learning, which is precisely what the authors have proposed in this work. Some of the limitations are:
1. Limited example/case-study: The authors solve one variant of pursuit-evade game called *Lady in the Lake*. The paper could be made stronger by showcasing superiority in other variants of differential games.
2. No significant contribution: While the authors show that their proposed method outperforms some of the RL algorithms when it comes to solving this game, I believe the contribution is not significant enough for ICLR. This is further substantiated by the fact that there exist similar physics-based (i.e., underlying governing equations are used as loss function to guide the training, instead of regressing with the ground-truth) methods for solving fixed-time nonzero-sum differential games [1] and similar zero-sum pursuit-evade game [2]. These are two representative papers which share flavor of this work ([1] is cited by the authors as well).

[1] Zhang, L. et al., *Pontryagin Neural Operator for Solving General-Sum Differential Games with Parametric State Constraints*. L4DC 2024.

[2] Bansal, S. et al., *DeepReach: A Deep Learning Approach to High-Dimensional Reachability*. ICRA 2021.

**Questions:**

Some comments and questions:

1. Perhaps it is a little misleading to say that the proposed method is not data-driven? My understanding is that to accurately learn the solution structure, you do need to ensure that you sufficiently cover the state-space during training.
2. How would the proposed method handle state-constraints, for e.g., collision?

---

### Official Review · Reviewer_t9HP · 2025-10-25

**Soundness:** 2
**Presentation:** 2
**Contribution:** 2
**Rating:** 4
**Confidence:** 4

**Summary:**

This paper explores an interesting problem (Lady in the Lake) for zero-sum games. The authors propose a learning-based framework that integrates the calculus of variations and the Pontryagin Maximum Principle (PMP) to learn saddle-point equilibrium strategies. This approach effectively mitigates the discontinuities in the value function that often arise from singular surfaces in conventional methods. Moreover, by jointly learning the state and costate dynamics, the method eliminates the need for manually designed heuristic reward functions commonly used in reinforcement learning. In the example of Lady in the Lake, the authors reformulate the system dynamics to facilitate learning through neural networks. Experimental results show that the proposed approach outperforms baseline methods and produces solutions consistent with the ground truth. Overall, the paper is clearly structured, well-organized, and complete.

**Strengths:**

The paper introduces a learning-based method to solve zero-sum differential games without requiring any supervised data. The authors provide clear and complete derivations, demonstrating how the Lady in the Lake problem is reformulated and how each learning module in their framework is designed. The mathematical formulations are correct and rigorous, and the overall presentation is complete.

**Weaknesses:**

The contribution of this paper appears limited due to its reliance on a single case study. Evaluating only the Lady in the Lake example is insufficient to convincingly demonstrate the effectiveness and generalization of the proposed SAL-DG framework. Although the authors present detailed and complete mathematical derivations, these analyses are closely tied to the specific problem setting. As a result, unless the authors can demonstrate that their proposed method can successfully solve other games, it is unclear whether the proposed method can be effectively extended to other zero-sum or general sum-difference games.

**Questions:**

1. There exisit some typos:
* In Eq. (1), it should be $\mu$ instead of $v_2$.
* In Eq. (4), it should be $u_1^*(t) = sgn(\theta(t_f))$, since $\lambda_{\theta}$ is constant and $\lambda_{\theta}(t_f)=sgn(\theta(t_f))$
* In Eq. (4), it should be $\sin u_2^*(t) = \frac{\mu}{r(t)}(\theta(t_f))$.

2. The contribution and purpose of Proposition A.2 are unclear. It appears to be redundant and is hard to catch up the authors' point.

3. In the ablation study, the authors state that “Figure 5 compares trajectories from the ground truth, SAL-DG, and …”. However, Figure 5 does not include the results for SAL-DG.

4. The paper mentions a control-based reward function used to train DDPG and TD3, but the design details are not provided. While the heuristic reward design is described clearly, the control-based formulation requires additional explanation.

5. Which conventional method the authors use to compute the ground truth?

---

### Official Review · Reviewer_U4KB · 2025-10-31

**Soundness:** 2
**Presentation:** 3
**Contribution:** 2
**Rating:** 4
**Confidence:** 2

**Summary:**

This paper proposes a framework "Structure-Aware Learning for Differential Games" (SAL-DG) for learning equilibrium strategies in certain pursuit-evasion differential games. The main idea of the framework is to encode the optimality conditions for equilibria (in this case, Pontryagin's Maximum Principle, the Hamilton-Jacobi-Isaacs equation, and terminal constraints) directly into a training objective which can be subsequently learned via neural networks. The authors mainly focus on applying this framework to the specific Lady-in-the-Lake pusruit-evasion example, which the authors show admits a certain objective reformulation amenable to end-to-end training. The authors evaluate experimentally the learned strategies of the SAL-DG method on the Lady-in-the-Lake example against two RL baselines.

**Strengths:**

The paper nicely motivates the desideratum of learning equilibrium strategies in differential games without data-driven approximations or reinforcement learning.

**Weaknesses:**

While the paper claims to introduce a general framework to learn equilibrium strategies for differential games, the paper restricts its focus solely on the Lady-in-the-Lake example game. Thus, it is not immediately clear how effectively the proposed approach extends to other differential games. To this end, the contribution of the paper seems quite limited.

Moreover, the paper is lacking in details related to the experimental evaluation of SAL-DG and the comparison methods. For example, for the experimental results in Section 5, there is no description of how the SAL-DG framework is actually implemented (e.g., the architecture of the control network), and the paper lacks (even a high-level description) of the RL baselines DDPG and TD3 that are used as comparisons. This lack of detail contributes more doubt as to whether the proposed method could generalize to other games beyond Lady-in-the-Lake.

**Questions:**

Q: As described under "Weaknesses", could the authors provide more details as to the training implementation of SAL-DG, as well as the comparison methods DDPG and TD3 in the experimental results?

Q: Is coordinate transormation and a reparameterization of the objective a necessary step (e.g., as in Proposition 4.1) for applying the SAL-DG framework to other differential games?

---

### Official Review · Reviewer_AU96 · 2025-11-06

**Soundness:** 3
**Presentation:** 3
**Contribution:** 2
**Rating:** 4
**Confidence:** 4

**Summary:**

The paper introduces Structure-Aware Learning for Differential Games (SAL-DG) -- a deep learning framework that solves zero-sum differential games by embedding analytical equilibrium conditions (from the calculus of variations and Pontryagin’s Maximum Principle, PMP) directly into the training objective, rather than relying on data, rewards, or expert supervision. The approach is validated on the Lady in the Lake pursuit-evasion problem, where it reportedly reproduces known analytical equilibria and outperforms reinforcement learning baselines such as DDPG and TD3.


The proposed method:

- jointly trains neural networks to represent the state, costate, and control trajectories.

- enforces PMP-based optimality conditions as loss functions, ensuring that learned trajectories satisfy necessary equilibrium constraints.

- uses a coordinate reparameterization (embedding angular variables as $(\sin \theta, \cos \theta )$) to remove discontinuities and stabilizes training.

- The framework optimizes variable terminal times and manifold-constrained terminal states, making it suitable for open-ended or geometry-constrained games.

**Strengths:**

- I like the theoretical grounding, in particular, the integration of PMP and calculus of variations into the training loss. To my knowledge, the inclusion of boundary and transversality conditions for variable terminal times is rare and nontrivial. While there is some prior work on free terminal time in ML/learning-for-control, the novelty of the paper is the combination with differential games, structure‐aware loss design, and terminal manifold constraints
- the *lady in the lake* benchmark is nice and appropriate
- The empirical results seem promising

**Weaknesses:**

## Positioning

While the paper’s slogan “Beyond Minimax” is conceptually appealing, it may be somewhat overstated. The formulation remains fundamentally a structured min–max optimization problem, where the key contributions lie in modeling - i.e., embedding PMP optimality and handling angular discontinuities. Once this setup is in place, the resulting optimization could, in principle, be solved by standard variational inequality methods (extragradient, optimistic GDA, etc.), which the paper neither compares against nor discusses. Clarifying this connection would strengthen the conceptual framing and highlight what is genuinely novel beyond the reparameterization and constraint embedding.


## On the generality of the mentohd  & real-world application

- While the authors argue the method is general, no evidence is provided that it scales beyond this toy domain (e.g., higher-dimensional systems or multi-agent interactions).

- The framework produces open-loop control trajectories, not feedback policies, limiting real-world applicability.

- It remains unclear how SAL-DG performs when the dynamics are noisy or partially unknown.

- While the proposed PMP-based training framework is in principle general to any differentiable dynamical system, Section 4.1’s state reparameterization assumes a single angular variable with circular topology. In many real-world systems, or even latent representations, the dynamics may evolve on higher-dimensional or non-Euclidean manifolds (e.g., spherical, toroidal, or group-structured spaces). In such cases, the proposed reparameterization and normalization trick would not apply directly, and it remains unclear how SAL-DG would handle manifold-valued states or hidden coordinate dependencies. Some discussion or generalization of this limitation would strengthen the paper’s applicability. In short, Section 4.1. is tailored specifically to the *Lady in the Lake* geometry


## Limited experiments & methods comparison

- All results are restricted to a single 2D pursuit–evasion game


No comparison to related “physics-informed” or “neural operator” baselines
- Recent methods like Pontryagin Neural Operator (Zhang et al., 2024) or Adjoint-Oriented Neural Networks (Yin et al., 2024) address similar principles of embedding PMP or Hamiltonian dynamics.
- A side-by-side comparison would clarify whether SAL-DG offers meaningful improvements in convergence, stability, or generalization.


## Theoretical Rigour: only necessary conditions enforced

SAL-DG enforces PMP-based necessary conditions but does not guarantee sufficiency or stability of equilibria.
The discussion acknowledges this, but the implications (e.g., sensitivity to initialization, possible spurious minima) are not empirically analyzed.



## Writing & Minor

Readers from the ML community may find it difficult to connect some terms used and the variational derivations to the implementation pipeline.

Abstract:
- The first sentence talks about learning without solutions or expert demonstrations, but the usual game setup is having agents' learning objectives; so it reads quite unclear what the paper is about.
- Similarly, the last sentence of the first paragraph talks about "such structures," but it's unclear to what that refers to

**Questions:**

1. How sensitive is SAL-DG to the choice of the heuristic time horizon $T$ used in the residual loss (Eq. 10)?

2. Have the authors tested the method on problems where PMP has no analytical solution (e.g., multi-evader games)?

3. Does enforcing PMP via residual minimization ever lead to inconsistent state-costate pairs (non-physical equilibria)?

4. Could the proposed approach handle general-sum differential games, where the equilibrium is Nash rather than a saddle-point?

5. How would SAL-DG behave under partial observability or unknown dynamics -- could it be combined with system identification?

6. How does the treatment of the transversality condition and costate boundary condition differs/improves upon previous “free time optimal control” ML methods?

7. What are some other real-world examples where the optimality conditions are known? How can performance be tracked in those cases?

8. I assume in some real-world examples, there might be hidden/latent spaces where the state dynamics evolve in polar coordinates (of even higher dimension than the one considered here). Is your method applicable in those cases?

---

### Author Response · Authors · 2025-12-02
**Rebuttal**

We thank the reviewers for their constructive feedback. We clarify the contribution and respond to specific points below.

***Clarification of novelty and contribution:***

While the use of optimality conditions in learning frameworks is not new, our contribution is novel in algorithmic design. First, our method integrates classical optimal control reasoning with deep learning for differential games, learning how to derive the optimal solution from necessary conditions mimicking what human scientists and engineers achieve. We explicitly follow the engineering workflow: (1) learn the control u(t) as a function of state and costate to minimize the Hamiltonian; (2) substitute this learned control into the PMP conditions to recover state and costate. This principled procedure distinguishes our framework from prior ML-based methods.
Second, our method enables the learning of both terminal time and terminal state through a calculus-of-variations formulation, making these quantities differentiable and optimizable within training. In contrast, existing approaches such as Pontryagin Neural Operator and Neural ODE-based frameworks require fixed time horizons for rollouts and therefore cannot support free-terminal-condition reasoning or optimization over terminal states.

***Justification of single-case study:***

We demonstrate this on a challenging instance with coupled dynamics, state constraints, and free terminal time, where SAL-DG converges while RL baselines fail. Although only one environment is shown due to space, the methodology is general; only the state parameterization is problem-dependent, not the framework. Neural networks naturally scale to higher-dimensional settings and non-analytic representations, further enhancing applicability beyond manual analytical methods.

***Comparison to baselines:***

We focus on DDPG and TD3 because physics-informed neural methods do not support free terminal time and terminal-state learning, and thus cannot be meaningfully applied to this setting.

***Responses to reviewers:***

Reviewer AU9606:

Sensitivity heuristic time horizon: The terminal time $t_f$ is a variable that is being learned end-to-end, not heuristic. Initialisation of $t_f$ affects convergence speed and possibly the outcome.

General-sum games: Yes. Each agent learns a control network to optimize its Hamiltonian; the framework extends naturally to Nash equilibria.

Partial observability / unknown dynamics: Yes. Unknown parameters can be included as learnable variables and optimized jointly (future extension).

Transversality and boundary conditions: Prior work fixes or heuristically tunes the terminal T. We learn T and the associated terminal conditions directly by gradient-based optimization.

Non-Euclidean latent systems: Reparameterization is standard engineering practice and enables learning under alternative representations. We show how we have to leverage both human prior knowledge and deep learning to solve problems.

Open-loop representation: In the derivation of saddle-point strategies, mathematicians typically first derive an open-loop representation of the saddle-point solution, and then use this open-loop formulation to synthesize the corresponding feedback saddle-point control (see Dynamic Noncooperative Game Theory, Başar)

Reviewer U4KB31:
Implementation details: All networks (state, costate, control) use 6-layer MLPs with constraint-handled activations. DDPG and TD3 use identical Q-network architectures for fairness.

Coordinate transformation: Reparameterization in Section 4.1 follows standard engineering practice to improve learning behavior. The core methodology in Section 4.2 is general and applies to a broad class of problems, for example, pursuit-evasion games in Cartesian coordinates can be handled without reparameterization, although in such cases an appropriate coordinate transform, as in Section 4.1, may yield better performance.


Reviewer t9HP25:
On Proposition A.2: It is not redundant; it provides a cleaner and computationally advantageous formulation of the optimality conditions employed by SAL-DG.

Reviewer tkW5:
On “not data-driven”: Our method does not require ground-truth optimal trajectories. Sampling the state space during training arises from solving the system itself, not from supervised data.
Handling state constraints: The presented experiment includes meaningful state and terminal constraints. Extending to nondifferentiable constraints (e.g., hard collisions) is future work.

---

### Meta-Review · Area_Chair_147Z · 2026-01-10

**Summary:**

The paper introduces a deep learning framework called Structure-Aware Learning for Differential Games (SAL-DG) that solves zero-sum differential games by embedding analytical equilibrium conditions (based on Pontryagin’s Maximum Principle (PMP) and the Hamilton-Jacobi equations) into the training objective. The paper also includes experiments on the game "Lady in the Lake" to illustrate the approach.

**Reviewer Concerns:**

There were a few concerns raised about the limitations and significance of the work. From theoretical point of view, only necessary conditions are provided for the success of the approach and the novelty is somewhat limited. From experimental point of view there was one toy case-study, without much evidence demonstrating the effectiveness of the approach and there was a lack of comparison with different methods. For those reasons, we recommend rejection.

**Reviewer Scores:**

The authors addressed some of the conceptual and technical questions of the reviewers, but the AC cannot forsee a significant change in the scores of the reviewers if they had been able to participate fully in the discussion, to make the paper pass the bar for acceptance in ICLR.

---

### Decision · Program_Chairs · 2026-01-26

Reject